# Human Milk Extracellular Vesicles: A Biological System with Clinical Implications

**DOI:** 10.3390/cells11152345

**Published:** 2022-07-30

**Authors:** Somchai Chutipongtanate, Ardythe L. Morrow, David S. Newburg

**Affiliations:** 1Department of Environmental and Public Health Sciences, University of Cincinnati College of Medicine, Cincinnati, OH 45267, USA; morrowa@ucmail.uc.edu; 2Division of Infectious Diseases, Department of Pediatrics, Cincinnati Children’s Hospital Medical Center, University of Cincinnati College of Medicine, Cincinnati, OH 45267, USA

**Keywords:** breastmilk, exosomes, extracellular vesicles, human milk, non-coding RNAs, maternal–child health outcomes

## Abstract

The consumption of human milk by a breastfeeding infant is associated with positive health outcomes, including lower risk of diarrheal disease, respiratory disease, otitis media, and in later life, less risk of chronic disease. These benefits may be mediated by antibodies, glycoproteins, glycolipids, oligosaccharides, and leukocytes. More recently, human milk extracellular vesicles (hMEVs) have been identified. HMEVs contain functional cargos, i.e., miRNAs and proteins, that may transmit information from the mother to promote infant growth and development. Maternal health conditions can influence hMEV composition. This review summarizes hMEV biogenesis and functional contents, reviews the functional evidence of hMEVs in the maternal–infant health relationship, and discusses challenges and opportunities in hMEV research.

## 1. Introduction

Extracellular vesicles (EVs; also known as exosomes) are cell-derived lipid bilayer submicron particles secreted from all types of mammalian cells into extracellular space. The primary function of EVs is to transport cellular components of the parent cells, including proteins, lipids, and nucleic acids, to recipient cells. They may elicit diverse complex biological processes within recipient cells, thereby influencing human physiology and pathology [1,2]. The discovery of vesicular transport machinery that governs vesicle trafficking from one cell and transfers cargos and elicits signaling in a recipient cell was so groundbreaking that it earned James Rothman, Randy Schekman, and Thomas Südhof the 2013 Nobel Prize in Physiology or Medicine [3]. EVs have been investigated to understand cell-to-cell communication and phenomena within the cellular microenvironment in various fields, including cancer biology [4,5], cardiology [6], coagulation [7,8], immunology [9], immunometabolism [10], neurology [11], and stem cell biology [12]. EVs released from specific cells have been studied for therapeutic purposes, including mesenchymal stem cell-derived EVs for regenerative medicine [13] and SARS-CoV-2 infection [14], and red blood cell-derived EVs for a drug delivery system [15]. EV molecular profiling has been investigated in clinically relevant biofluids, e.g., plasma [16], urine [17], cerebrospinal fluid [18], amniotic fluid [19], and saliva [20] as candidate biomarkers of disease diagnosis or prognosis.

Human milk, a complex and dynamic biofluid, contains nutrients that support infant growth as well as bioactive components that protect infants against various diseases [21,22,23,24]. Clinical and epidemiologic studies confirm the beneficial effects of feeding human milk over infant formula in preventing early and long-term diseases, e.g., necrotizing enterocolitis, neonatal sepsis, respiratory and gastrointestinal tract infections, allergic diseases, obesity, diabetes mellitus, and malignancies [21,22,23,24]. Knowledge regarding mechanisms by which human milk components deliver positive health outcomes to children and young adults is growing. The recognized human milk bioactive components include proteins (immunoglobulins, lactoferrin), growth factors, cytokines, adipokines, non-digestible oligosaccharides (2′-fucosyllactose (2′FL), lacto-N-tetraose (LNT), lacto-N-neotetraose (LNnT), sialyllactoses (3SL, 6SL)), leukocytes, and stem cells [25,26,27,28]. In 2007, Admyr et al. [29] reported that human milk contains EVs harboring major histocompatibility complex (MHC) class I/II, which can be immunosuppressive. Human milk extracellular vesicles (hMEVs) are now considered a functional component of human milk, and further elucidation of this biological system could provide a unique opportunity to study maternal-to-child biochemical communication with intergeneration health consequences.

Searching the PubMed database for (“human milk” OR breastmilk) AND (exosomes OR “extracellular vesicle”) yields 100 articles since 2007 with the majority published over the last five years (Figure 1). This increasing appreciation of the potential roles of hMEVs also suggests there are many unknown functions of hMEVs to be explored further. This review summarizes the known components of the hMEV biological system, including cell sources, vesicular biogenesis, subpopulations, and molecular composition. How these components interact with maternal conditions, and their potential biological influence on neonatal and infant growth and health, is of particular interest. Opportunities and challenges of future hMEV research include potential clinical applications of hMEV-based biomarkers to predict maternal–child health outcomes and hMEV-based therapy.

## 2. Biology of hMEVs

### 2.1. Biogenesis and Subpopulations

Extracellular vesicle (EV) is a generic term covering three vesicle subpopulations: exosomes, microvesicles, and apoptotic bodies. While these EV subpopulations share the same plasma membrane and cytosolic components of the parent cells, they are different in intracellular origin, biogenesis, and release mechanisms, which results in various vesicular sizes and compositions [30,31].

Exosomes (approximately 40–150 nm) originate from the inward budding of endosomal membrane into intraluminal vesicles (ILVs) from which are generated multivesicular bodies (MVBs), which are transported to and fuse with the plasma membrane to be released as exosomes into the extracellular space [32,33] (Figure 2). The generation of multivesicular bodies is mediated by at least two distinct pathways and involves sorting of various molecules into intraluminal vesicles. The first pathway utilizes the Endosomal Sorting Complex Required for Transport (ESCRT). This machinery contains up to 30 proteins which can be divided into four protein complexes: ESCRT-0, -I, -II, -III, and the associated ATPase Vps4 complex [34,35,36,37]. ESCRT-0 recognizes and sorts the ubiquitinated cargo proteins into the lipid domain [38,39]. ESCRT-I and -II invaginate the late endosomal membrane to form buds with sorted cargos [40,41]. ESCRT-III de-ubiquitinates the protein cargo [42,43]. After recruiting the Vps4 complex to fully assemble the ESCRT-III-Vps4 scission machinery, this complex catalyzes vesicle abscission to produce the ILVs that form MVBs [44,45,46]. Vps4 also plays roles in ESCRT disassembly and recycling of subunits for further rounds of vesicle formation [47,48]. The second pathway of MVB formation is ESCRT-independent [32,33,49]. In this mechanism, MVB is generated from raft-based microdomains of endosomal compartments where the neutral sphingomylinase 2 (nSMase2) converts sphingolipids into ceramide [50]. This sphingolipid-to-ceramide conversion induces coalescence of the microdomains into larger structures which then promotes domain-induced budding and formations of ILVs and MVBs [32,49]. Inhibition of nSMase2 has been shown to reduce the release of exosomes in several types of cells, but not all, suggesting that the role of ceramide in exosome release varies among cell types [33,49,50,51,52,53,54]. 

Unlike exosomes, microvesicles (usually 100–1000 nm in size) originate from the direct outward blebbing and pinching of the plasma membrane in a constitutive manner or upon stimulation [32,55]. Cytoplasmic protrusions are due to rearrangement of plasma membrane asymmetry induced by the Ca^+2^-dependent enzymes flippase (aminophospholipid translocase) and floppase (ABC transporter). Flippase translocates phosphatidylserine and other phospholipids from the inner leaflet to the outer leaflet; conversely, floppase translocates the phospholipids from the outer leaflet to the inner leaflet [55,56,57,58,59]. Cargo sorting and microvesicle shedding are tightly regulated by several small GTPases, including ARF6, Rab, Rac1, and RhoA or by modification of the lateral pressure of phospholipids via phosphatidylserine binding protein on the inner leaflet, or sphingomyelin/cholesterol binding proteins on the outer leaflet [55,60,61,62].

Apoptotic bodies, typically 1000–5000 nm, are released from dying apoptotic cells by the outward blebbing of the plasma membrane. This process is regulated by caspase-cleaved substrates such as Rho-associated kinase 1, plexin B2, and pannexin 1, which phosphorylate and activate the myosin light-chain that regulates blebbing [63,64]. Apoptotic bodies are usually packed with DNA fragments, proteins, and parts of cellular organelles. Apoptotic bodies are often underappreciated in the field of EV research. Apoptotic bodies may play roles as messengers from dying cells to regulate various cellular processes, such as cell clearance and homeostasis, pathogen dissemination, and immune responses [64].

The complexity of EVs does not allow them to be fully characterized by any single method; several biophysical and biochemical methods are required. To promote reproducibility of EV studies and enhance ability to interpret findings across studies, the International Society of Extracellular Vesicles has recommended the minimal information needed to define a preparation as extracellular vesicles (MISEV) in 2014 [30], with an update in 2018 [31]. Nanoscale-vesicle morphology and particle size distribution in tandem with enrichment of multiple exosomal biomarkers and depletion of common protein contaminants are required to claim the presence of exosomes in isolates [30,31]. Therefore, consistent with the consensus recommendation of the International Society of Extracellular Vesicles, the term extracellular vesicles or EVs are used instead of exosomes throughout this review with only a few exceptions [30,31]. The intent is to avoid the confusion in previous literature due to labeling several types of EV preparations as exosomes.

### 2.2. Cell Sources

Current evidence suggests that hMEVs are a mixed population of vesicles released from the local breast tissues and distant organ compartments [29,65,66]. HMEVs are mainly produced and secreted from mammary gland epithelial cells during lactation [29,65], some of which are attached to the surface of human milk fat globules (HMFGs). Cells in human milk, e.g., lymphocytes, macrophages, and stem cells, may also contribute to the EVs presented in human milk; in addition, EVs from other organ systems that transmigrate to milk through the systemic circulation can also contribute to the pool of EVs in human milk [26,27,29,66]. While the relative contribution of local and distant organ systems would define the characteristics and functions of hMEVs, the exact proportion has not been determined due to technical limitations. Advances in single EV technologies in which isolation and characterization of individual single-particle EVs [67,68,69,70] would facilitate the definition of source-specific hMEVs as part of the mixture of EVs in the human milk biological system.

### 2.3. Molecular Components

HMEVs are composed of proteins, nucleic acids, and lipids derived from plasma membrane and cytoplasm of the cells of origin (Figure 2). However, the functional cargos are not components of parent cells passively incorporated into hMEVs, but rather depend on active sorting mechanisms [71]. By high-throughput analyses, up to 920 proteins [72], 1523 miRNAs [73], and 395 lipids [74,75] have been identified in hMEVs. The expression and amount of hMEV molecular components vary depending on the maternal physiological and pathological states [76,77,78,79,80,81,82]. Nonetheless, some of the functional molecules are highly abundant and frequently identified in hMEV studies. These high abundance proteins and non-coding RNAs (ncRNAs) of hMEVs and their putative functions are summarized in Table 1.

The proteins tetraspanins (e.g., CD9, CD63, CD81), Tsg101, and Alix play crucial roles in the exosome biogenesis and are usually measured as the biomarkers of exosome enrichment during hMEV isolation and characterization. Lactadherin (or Milk Fat Globule-EGF-factor VIII; MFGE8), butyrophilin, and xanthine dehydrogenase/oxidase, major proteins of milk fat globules, are highly abundant in hMEVs and often serve as specific milk EV markers [90,91]. Butyrophilin, together with major histocompatibility complex (MHC) and transforming growth factor-β (TGF-β), may work in concert to support the immunomodulatory properties of hMEVs [29,94,96]. Furthermore, Butyrophilin and TGF-β in hMEVs may be responsible for the induction and differentiation of CD4^+^CD25^+^FoxP3^+^ T regulatory cells from peripheral blood mononuclear cells [29,85,96]. Mucin-1 expressed on hMEVs could bind DC-SIGN on monocyte-derived dendritic cells to inhibit HIV-1 viral transfer to CD4+T cells [100]. Adhesion molecules, i.e., intercellular adhesion molecule-1 (ICAM-1) and integrins, are on the hMEV surface and play roles in vesicular trafficking and recipient cell targeting [83,86]. Lactadherin, a milk glycoprotein that protects against symptomatic rotavirus infection [92], is also expressed on hMEVs [29,90,91]. Lactadherin may be responsible for the anti-rotavirus effect of hMEVs in vitro [132].

MicroRNAs (miRNAs) are small ncRNA molecules (~22 nucleotides) involved in epigenetic regulation and pre-transcriptional gene repression. Most miRNAs in hMEVs are derived from mammalian epithelial cells [66] and play critical roles in epigenetic programming of intestinal homeostasis, immunomodulation, metabolic regulation, and neurodevelopment during the postnatal period [73,105,133,134,135]. Recently, Ting et al. [103] conducted a systematic review on miRNAs in human milk and reported miR-148a-3p, miR-30a/d-5p, let-7a/b/f-5p, miR-22-3p, miR-146b-5p, and miR-200a/c-3p to be the top 10 highly abundant miRNAs in all human milk fractions including hMEVs. Note that miR-148a-3p is the most abundant miRNA in hMEVs, and is involved in multiple cellular processes, including regulating DNMT1-dependent DNA methylation [105,106,107], suppressing tumor growth and metastasis [106,107,108,109,110,111], mitigating NF-κB mediated intestinal inflammation [112], modulating angiogenesis [113,114], and exerting neuroprotective effect [115,116,117,118,119]. Thus, hMEV miRNAs exhibit a wide array of functions in both normal biology and disease, and warrant further study.

The EV membrane contains the proteins discussed above embedded in a lipid bilayer originating from the plasma membrane of parent cells; therefore, the major lipids are cholesterol, sphingomyelin, phospholipids, and ceramides [74]. In addition, lipidomic analysis identified 395 lipids in hMEVs isolated from term and preterm breastmilk [75]. The 50 lipids at greatest concentration are associated with the ERK/MAPK signaling pathway and intestinal cell regulation. Moreover, the molecular species of phosphatidylethanolamine PE(18:1/18:1), phosphatidylcholine PC(18:0/18:2), and PC(18:1/16:0), and phosphatidylserine PS(18:0/18:1) and PS(18:0/22:6) are enriched in hMEVs derived from both preterm and term milk; these species could support neurodevelopment and long-term health outcomes [75].

Human milk oligosaccharides (hMOS), the third most abundant component of human milk, inhibit binding of pathogens to the intestinal mucosa, stimulate growth of mutualist microbes in the gut (prebiotic), and modulate signaling and inflammation in the intestinal mucosa [28]. Using liquid chromatography-mass spectrometry, He et al. discovered that hMOS are also encapsulated by hMEVs from both colostrum and mature milk. For colostrum hMEVs, 2’-fucosyllactose (2’FL), lacto-N-tetraose/neotetraose (LNT/LNnT), lacto-N-difucohexaose (LDFH) were predominant; for mature milk hMEVs, 2’FL, lacto-N-fucopentaose I (LNFP I), and LNT/LNnT predominated [136]. 2’FL exhibits immunomodulatory and anti-infective properties [28,136,137,138,139,140,141]. In adherent invasive *E. coli*-infected mice, hMEV-encapsulated hMOS could attenuate intestinal inflammation comparable to or perhaps superior to that of free solutions of 2’FL [136]. Such research into the full potential of hMOS as part of hMEV functional cargos is a promising area of research.

A breakthrough study led by Flynn et al. revealed that some non-coding RNAs are modified by N-glycans and expressed on the outer surface of mammalian cells, i.e., cell-surface glycoRNAs [142]. Cell-surface glycoRNAs can interact with sialic-acid-binding immunoglobulin-like lectins (Siglecs) [142,143]. Thus, glycosylation of RNAs could help regulate cellular physiology and innate immunity. As EV plasma membranes originate from the parent cells, glycoRNAs could well be present on the surface of EVs as well, including hMEVs. GlycoRNA biogenesis remains opaque [144], and EV-surface glycoRNAs seems to be a rich research topic for understanding human milk as a biological system.

### 2.4. Bioavailability and Tissue Distribution

From a physiologic standpoint, a primary issue of hMEVs is their stability upon ingestion and during the gastrointestinal digestion of milk. HMEVs survive under simulated gastric/pancreatic digestion for at least one hour [145,146]. Using simulated gastric/pancreatic digestion in vitro, Liao et al. [146] demonstrated that the SDS-PAGE protein profiles were similar between undigested and digested hMEVs, and analysis by protein array indicates that Alix, ICAM-1, and flotillin-1 were intact in the digested hMEVs. Fluorophore-labeled digested hMEVs could be internalized by human intestinal epithelial crypt-like cells (HIEC) to the same degree as undigested vesicles, with approximately 10% of the internalized hMEVs located at the nucleus [146]. These findings suggest that the lipid bilayer of hMEVs enclose and protect their molecular cargos against degradation within the gastrointestinal tract.

A related physiologic question is the bioavailability and tissue distribution of hMEVs following oral consumption. In vitro, human vascular endothelial cells internalize bovine milk-derived EVs; this is a crucial step for the delivery of dietary EVs and their functional cargos to systemic circulation and peripheral tissues [147]. The rate of EV uptake by human vascular endothelial cells was decreased approximately 50% by removal of bovine milk-derived EV surface proteins by proteinase K or by the presence of D-galactose, a competitor of cell surface carbohydrate binding; these data suggest a significant role of glycoproteins in milk EV transmigration. [147]. In vivo, fluorophore-labeled bovine milk-derived EVs, administered to C57BL/6 mice retro-orbitally, transmigrate across vessels for delivery to tissues, mainly in the liver and spleen, with trace amounts detected in the stomach, intestines, and lungs [147]. Relative to intravenous administration in mice, oral gavage of fluorophore-labeled bovine milk-derived EVs showed an apparent bioavailability of 4% and 6% at 3 h and 24 h, respectively [148]. At 24 h, most intravenously administered bovine milk-derived EVs accumulated in liver, spleen, and brain of Balb/c mice, while tissue distribution of orally administered bovine milk-derived EVs (higher to lower) expanded to liver, spleen, kidneys, heart, lungs, and brain [148]. In addition, the distinct miRNA cargos of the orally administered bovine milk-derived EVs demonstrated their unique tissue distribution patterns [148]. Transfecting synthetic fluorophore-labeled miRNAs into bovine milk-derived EVs followed by oral administration resulted in tissue distribution patterns at 24 h (higher to lower) of miRNA-320a: liver, spleen, and kidneys; and of miR-34a and miR-155-5p: brain and spleen [148]. These data highlight the potential bioavailability and tissue selectivity of bovine milk-derived EVs and their miRNA cargos. Future studies are warranted to elucidate tissue distribution patterns following hMEV administration. Direct investigation of the underlying mechanisms should yield information of high utility in understanding the potential impact of hMEVs on both local and systemic physiology.

### 2.5. Biological Functions

HMEVs are heterogenous due to different cell sources and altered compositions during maternal health and disease. The net effect on target cells results from concerted actions of the sum of functional hMEV cargos delivered rather than from a single hMEV component [149]. Furthermore, a dose-response relationship will favor activities of highly abundant cargos (as shown in Table 1) relative to low abundance molecular components of hMEVs. Despite the dearth of data on the relationship between whole vesicle bioactivities and each hMEV functional moiety, current understanding supports several strong putative biological impacts: HMEVs promote gut maturation and homeostasis [133,149,150,151,152,153]. HMEVs can attenuate mucosal inflammation and perform other forms of immunomodulation [29,149,154]. HMEVs exhibit antiviral effects against HIV-1, rotavirus, respiratory syncytial virus (RSV), human cytomegalovirus (CMV), Zika virus, and Usutu virus [100,132,155,156]. Reported hMEV functions are summarized in Table 2.

Direct effects of hMEVs on neurodevelopment remain to be demonstrated. However, promising associations were observed between miR-148a-3p (the most abundant hMEV-derived miRNA) and neuroprotection in Alzheimer’s Disease [115,116], ischemic stroke [117], and temporal lobe epilepsy [118]. In both term and preterm infants, human milk feeding is associated with improved neurodevelopmental outcomes [157,158]. This has been partially attributed to hMOS, including 2’FL, 3’-sialyllactose (3’SL) and 6’-sialyllactose (6’SL), for their contribution to synaptic formation, neuro-transmission, and memory improvement [159,160,161,162,163,164,165]. HMEV-derived miR-148a-3p may work in concert with hMOS and other human milk components on brain development during infancy.

## 3. Maternal Conditions Influence hMEV Composition and Child Health Outcomes

Maternal health conditions impact the content of nutrients and bioactive components of human milk [166]. Whereas hMEVs transfer functional molecules from mothers to growing infants, studying alterations of hMEV functional cargos affected by maternal factors could serve as a key to understand intergenerational health consequences [76,77,79,82].

### 3.1. Maternal Stress

Maternal stress/psychological distress is associated with negative child health outcomes, including poor nutritional status [167], reduced linear growth [168], altered neurodevelopment [169,170], and increased risk of childhood asthma and atopic diseases [171,172]. In a cohort of 80 mothers, Bozack et al. [82] evaluated the association between maternal lifetime stress, including negative life events during pregnancy, and hMEV-derived miRNAs (hMEV-miRs). Among 205 hMEV-miRs, increased expression of six miRNAs, including miR-99b-3p, miR-96-5p, miR-550a-5p, miR-616-5p, miR-155-5p, and miR-604, were significantly associated with the measures of maternal stress [82]. These differentially expressed hMEV-miRs may be involved in epigenetic regulation of fatty acid metabolism, steroid biosynthesis, and the Hippo signaling pathway that regulates organ growth [82]. Dysregulation of the Hippo pathway is associated with metabolic diseases, e.g., obesity, diabetes, fatty liver, and cardiovascular disorders [82,173], and atopic diseases, including asthma [174,175]. Measuring direct biological impacts of maternal stress-induced hMEV-miR changes on early-life programming and long-term health outcomes in breastfed infants could lead to better understanding of one of the mechanisms of the long recognized but poorly understood connection between maternal stress and infant outcome.

### 3.2. Maternal Overweight/Obesity

Maternal overweight and obesity have impacts on human milk macronutrients and bioactive molecules with the potential to increase the long-term risk of child obesity and impaired neurodevelopment [176,177,178,179,180]. Overweight/obese mothers had lower hMEV-derived miR-148a and miR-30b at 1-month of lactation (30 normal weight vs. 30 overweight/obese). After controlling for gestational age, gender, and birth weight, both miR-148a and miR-30b intake were significantly associated with infant anthropometric measures [79]; for each fold decline in hMEV-derived miR-148a, the infant body weight and fat mass were increased by 0.6 kg and 0.3 kg, respectively. The significance of the relationship between hMEV-derived miR-148a and infant anthropometric measures diminished at 3 to 6 months of lactation [79]. MiR-148a is the precursor of miR-148a-3p, the most abundant miRNA in hMEVs, which has known neuroprotective and neuro-proliferative effects [103,115,116,117,118]. Accordingly, the reduction of hMEV-derived miR-148a in maternal overweight/obesity provides some mechanistic insight into increased risk of childhood obesity and unfavorable neurodevelopmental outcomes in obese mothers.

### 3.3. Maternal Diabetes

Type I diabetes is an autoimmune disease whose onset is typically in childhood. Children who were breastfed had half the risk of type 1 diabetes as those fed infant formula [181]. Mirza et al. [77] demonstrated that hMEVs of mothers with type 1 diabetes were enriched with immune-modulating miRNAs relative to healthy mothers (n = 26 in each group). Of the 631 identified miRNAs, 9 hMEV-miRs were significantly altered (6 upregulated and 3 downregulated) in mothers with type 1 diabetes [77]. These 9 hMEV-miRs are involved in cell cycle regulation and immune response processes, and included PI3K/AKT mediated proinflammatory cytokine production. Two miRNA mimics of miR-4497 and miR-3178, which are significantly upregulated in hMEVs from type 1 diabetic mothers, enhance the release of TNF-α pro-inflammatory cytokine in vitro from transfected THP1 monocytes [77]. However, hMEV-miRs from mothers with type I diabetes did not increase the risk of type I diabetes or other inflammatory diseases in their offspring.

Gestational diabetes mellitus (GDM) affected the miRNA cargos of hMEVs [78]. In a cohort of 32 GDM and 62 non-GDM, the levels of HMEV-miRs involved with metabolism (e.g., miR-148a, miR-30b, let-7a, and let-7d) were measured in milk along with infant growth and body composition in the first six months of life [78]. Levels of miR-148a, miR-30b, let-7a, and let-7d were lower in GDM human milk. MiR-148a were negatively associated, while levels of miR-30b were positively associated, to infant weight and fat mass at 1 month of age [78]. Thus, gestational diabetes mellitus mothers produced aberrant hMEV-miR levels, which were associated with abnormal metabolic outcomes in their nursing offspring.

### 3.4. Premature Delivery

Preterm delivery is associated with differences in functional cargos of hMEVs. Mourtzi et al. [80] compared hMEVs and their lncRNA levels in mothers of term (≥37 weeks of gestation) vs. preterm (<37 weeks of gestation) birth (n = 10 each group). Of the 31 hMEV-derived lncRNAs measured, 4 lncRNAs differed significantly in milks of the preterm group; LRRC75A-AS1 was higher, and CTC-444N24.11, CRNDE and LINC00657 (NORAD; non-coding RNA activated at DNA damage) were lower. LRRC75A-AS1 or SNHG29, involved in accelerating cellular senescence and triggering pro-inflammatory cytokine production, was also higher in preterm birth placentas [182]. The function of CTC-444N24.11 is unknown. CRNDE is a metabolic regulator that promotes aerobic glycolysis [183], while LINC00657 helps maintain genomic stability under stress [184,185]. Alterations of these lncRNAs in hMEVs are consistent with adaptive responses of the preterm infants against hypoxic conditions [80]. In addition, LINC00657 is one of the most abundant lncRNAs in hMEVs, and its decrease may serve as a biomarker of perinatal hypoxic stress [80].

The peptides present in hMEVs from mothers with term and preterm birth were compared by proteomics; 719 peptides were identified [81]. Differential expression and bioinformatic analyses revealed 70 peptides whose levels differed between groups, with 47 being higher in the term mothers, and 23 being lower. The biologic activities of these peptides mainly involve cell proliferation and development, biological adhesion, immune responses, and metabolic process [81]. These biologically active peptides included lactotransferrin (LTF)-derived peptide residues 79–96 (DGGFIYEAGLAPYKLRPV, transferrin-like 1 domain), and lactadherin (MFGE8)-derived peptide residues 24–47 (LDICSKNPCHNGGLCEEISQEVR, EGF-like domain) [81]. The LTF (residues 79–96) peptide probably has anti-inflammatory and antimicrobial properties, while the MFGE8 (24–47) peptide may exhibit cell growth promoting effects [81]. The preterm hMEVs induced greater cell proliferation and wound healing on FHC human intestinal cells compared to term hMEVs [81]. In a necrotizing enterocolitis (NEC) animal model, the oral administration of the preterm hMEVs protected the villous integrity from injury and restored enterocyte proliferation relative to untreated NEC-like mice [81]. This structure–function relationship suggests that mothers with preterm birth secrete hMEVs containing factors that protect against the consequences of the immature gut in their infants.

### 3.5. Maternal Allergic Sensitivity and Lifestyle

Levels of immunologic components of human milk, including hMEVs, are associated with maternal allergies and environmental factors [76,186,187,188]. Torregrosa Paredes et al. [76] characterized the influence of maternal allergic sensitivity and anthroposophic lifestyle on hMEV-derived protein levels and their relationship with the child sensitivity at 2 years of age. Two hMEV subpopulations were evaluated: HLA-DR-enriched hMEVs and CD63-enriched hMEVs. In the HLA-DR-enriched hMEV subpopulation, significantly lower levels of mucin-1 were detected in mothers with anthroposophic lifestyle compared to non-anthroposophic mothers [76]. In the CD63-enriched hMEV subpopulation, lower levels of mucin-1 were detected in sensitized mothers, while higher levels of HLA-ABC were associated with mothers whose children developed sensitization (allergen-specific IgE levels ≥0.35 kUA/L) [76]. This study highlights the complex interaction of maternal underlying conditions and environmental factors on hMEV composition and child health outcomes.

Table 3 summarizes current knowledge on hMEV molecular and functional changes affected by maternal health and disease states. These sets of significant proteins and miRNAs also hold promise as predictive biomarkers of child health outcomes. 

## 4. Challenges and Clinical Implications

HMEVs, like EVs in other biofluids and from stem cells, have promising potential clinical applications. However, several characteristics of hMEVs are distinct from other EVs. HMEVs are expressed as part of human breastmilk, harbor selective molecular packages of parent cells found primarily in the mammary gland and elicit epigenetic regulation and cell signaling in infant tissues. Thus, hMEVs represent another mechanism whereby human milk acts as a medium for communication from the mother to the infant, and current data suggest that this may be one of the primary effectors of this vertical communication. This review focuses on the EVs in human milk, but EVs undoubtedly have analogous functions in other mammals.

HMEVs are a complex biological system of human milk. They originate in parent cells in the local breast tissues and, to a lesser degree, distant organ systems. After the infants ingests the milk, their intestinal mucosa may be a primary target, but appreciable amounts of hMEVs are also absorbed into the infant circulation. The membrane contains adhesion molecules that can guide the vesicle to specific types of cells, both in the gut lining, and the distant organ systems after some of the hMEVs pass into the blood circulation [83,86,147,189,190,191]. The cargo of the hMEVs include both biologically active proteins and miRNAs that are capable of programing the recipient cells to induce physiological responses in breastfed infants. Understanding more about this system of mother to infant communication could serve two major clinical applications: predictive biomarkers and therapeutic agents (Figure 3).

A common problem in searching for biomarkers is reproducibility across populations [192,193]. Several strategies help address this challenge. First, a large sample size during the discovery phase of biomarker research (e.g., 100s–1000s) combined with validation using a large-scale independent cohort or multi-center study would enhance the probability of finding valid and robust population-based biomarkers [192,193,194]. Second, a mechanistic biomarker that is directly involved in physio-pathological processes is a strong candidate [192,195]. The mechanistic marker would be improved further if it included multiplex biomarkers; measuring several markers in combination is likely to be more specific than a solitary descriptive biomarker [17,196,197]. While a population-based biomarker strategy is often constrained by budget, mechanistic/multiplex biomarkers may be more compatible with hMEVs functional proteins and miRNAs. The ideal is to incorporate both strategies in the form of hMEV-based (mechanistic/multiplex) biomarker studies conducted as an extension of population-based longitudinal birth cohorts, e.g., Boston (ClinicalTrials.gov Identifier NCT03228875), Influenza IMPRINT [198], MatCH [199], NEHO [200], NICE [201], and PREVAIL [202]. This would provide the strongest opportunity to discover hMEV-based biomarkers for predicting maternal–child health outcomes.

To develop hMEVs for therapeutic purposes, a major challenge is the quantity and quality of hMEVs available for research. In general, a single-step EV isolation method, i.e., high-speed ultracentrifugation without a prior step of low-speed centrifugation, polymer precipitation, ultrafiltration, or size-exclusion chromatography, provides high-to-intermediate yields, but such isolates may be contaminated with lipoproteins, or free and aggregated proteins [31]. A combination of isolation methods such as differential ultracentrifugation coupled with polymer precipitation or size-exclusion chromatography achieve a higher specificity with less contaminates, but usually with a lower recovery [31,203,204]. There is no single optimum/gold-standard method for EV isolation [31]. The isolation method must be chosen to consider compatibility and suitability to downstream applications. For example, a single-step method may be superior for large-scale hMEV biomarker studies, while combined methods may be more suitable to prepare hMEVs for therapeutic testing. A comparison of different methods for pre-processing and isolation of hMEVs [205] concluded: (i) milk fat should be removed prior to long-term milk storage; (ii) different combinations of isolation methods (differential ultracentrifugation plus serial filtration vs. precipitation (i.e., ExoQuick) plus serial filtration) provide comparable yields; (iii) combined methods were effective for hMEV isolation even with a small starting volume of 1.5 mL human milk [205].

As one of the human milk bioactive components, hMEVs are expected to be safe for oral consumption, and without toxicity, supporting its use as a nutraceutical/therapeutic agent. Perhaps one potential use of isolated hMEVs would be human milk fortification [206,207] to enhance the beneficial effects of maternal breastmilk or donor milk or to improve infant formula. For preterm infants, especially whose with very low birth weight, this could help reduce their high risk of morbidity and mortality [207,208]. Another application for isolated hMEVs would be as a therapeutic agent to mitigate intestinal inflammation and facilitate tissue repair in preterm infants with NEC [81,112]. HMEVs also harbor functional molecules with anti-tumor and anti-viral effects, e.g., lactadherin which can inhibit rotavirus [92] and miR-148a-3p which inhibits gastric and pancreatic cancers [109,110]. The therapeutic potential of bovine milk-derived EVs, including those EVs loaded with miR-148a-3p, have been investigated in colon cancer [209,210]. Future preclinical and clinical studies should evaluate the safety and efficacy of hMEV-based therapy against NEC, rotavirus infection, and adult gastrointestinal cancers.

A major hurdle for use of hMEV for therapeutics is the amount of human milk available for isolating hMEVs. Although human milk banks have large quantities of donor milk, it is mostly reserved for neonates in immediate need. Moreover, the stability of hMEVs to the processing and storage conditions of donor human milk requires investigation [211,212].

The synthesis of hMEV-inspired therapeutic nanoparticles is an alternative strategy for harnessing therapeutic potential of hMEVs. For example, synthetic EVs could be nanoparticle platforms for drug and gene delivery [15,213]. Targeted oral delivery by bovine milk EVs and hMEV-based carriers have been actively investigated to overcome the solubility limits and toxicity of the chemotherapeutic drug paclitaxel [214,215]. EVs also improve the oral bioavailability of small molecules, such as celastrol [216], anthocyanidins [217,218], and curcumin [219,220,221]. The stability and targeted delivery of small interfering RNAs (siRNAs) is also improved by using EV carriers [222,223,224]. SiRNA-loaded milk EVs are safe and effective in vivo [224,225].

The use of synthetic mimetics of hMEV-derived functional molecules can overcome the limited availability of human milk, errors during EV isolation and characterization, and also avoid any undesirable effects by unknown or unwanted EV components [226,227]. HMEV-inspired therapeutic liposomes incorporating desirable protein ligands and miRNAs (Table 2) would be amenable to standardization and large-scale clinical application. Quantitative engineering using stepwise quantitative loading demonstrated that the bottom-up synthetic fibroblast-liked EV mimics had precisely controlled lipid (43% cholesterol, 16% sphingomyelin, 38% phospholipids, 2% phosphatidic acid, 1% diacylglycerol), protein (CD9, CD63, CD81), and miRNA (miR-21, miR-124, miR-125, miR-126, miR-130, miR-132) components. These synthetic EV mimetics promoted wound healing and neovascularization in 2D and 3D in vitro models [227], akin to the known therapeutic potential of fibroblast-derived EVs on diabetic wound healing [228]. This synthetic bottom-up synthesis promises to harness the full potential, designability, and therapeutic effects of hMEV-like EV mimics in pediatric and adult populations in the future.

## 5. Conclusions

As a unique biological system of human milk, hMEVs deliver functional cargos of proteins, nucleic acids, and lipids from mothers to breastfeeding infants. These cargos induce epigenetic and physiological responses in growing infants, including accelerating gut maturation, attenuating mucosal inflammation, modulating the immune system, and preventing viral infection. The variation in hMEVs among mothers of diverse metabolic states provide special opportunities in searching for predictive biomarkers of child health outcomes, particularly for those that also may serve as mechanistic markers. Some of the biomarkers may also be relevant to broader applications in adult medicine. Therapeutic hMEVs, both natural isolates and inspired synthetics, hold great promise as novel biologics for human disease, and warrant expansion of our research efforts.

## Figures and Tables

**Figure 1 cells-11-02345-f001:**
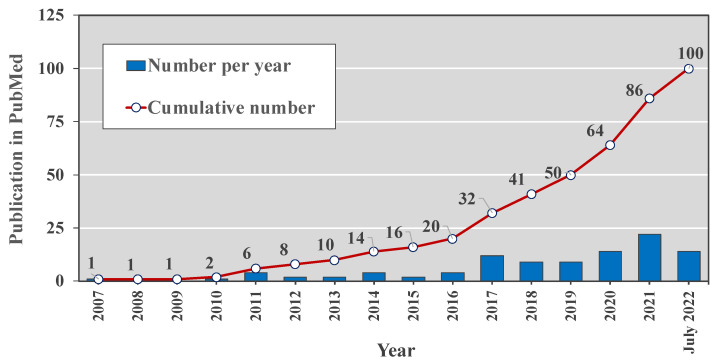
The number of peer-reviewed publications in the PubMed database during 2007–2022 with search terms (“human milk” OR “breastmilk”) AND (“exosomes” OR “extracellular vesicle”).

**Figure 2 cells-11-02345-f002:**
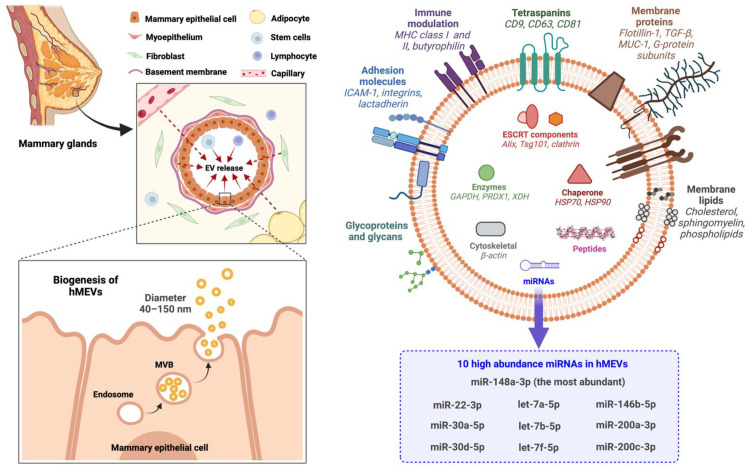
HMEV biogenesis and compositions. EV, extracellular vesicles; GAPDH, glyceraldehyde-3-phosphate dehydrogenase; hMEVs, human milk extracellular vesicles; HSP, heat shock protein; ICAM-1, intercellular adhesion molecule 1; MHC, major histocompatibility complex; miRNAs, microRNAs; MUC-1, mucin-1; MVB, multivesicular bodies; PRDX1, peroxiredoxin 1; TGF-β, tumor necrosis factor-β; Tsg101, tumor susceptibility gene 101; XDH, xanthine dehydrogenase.

**Figure 3 cells-11-02345-f003:**
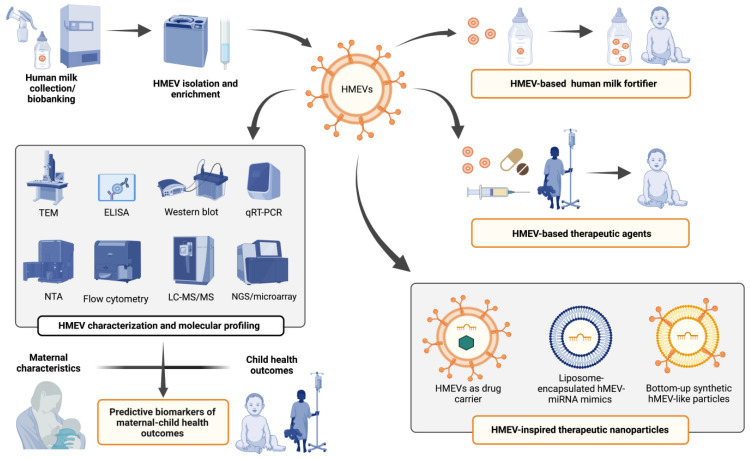
Potential clinical applications of hMEVs as predictive biomarkers of maternal–child health outcomes and as a source or inspiration for novel nutraceuticals/therapeutics. Abbreviations: TEM, transmission electron microscopy, ELISA, enzyme-linked immunosorbent assay; qRT-PCR, quantitative real-time polymerase chain reaction; NTA, nanoparticle tracking analysis; LC-MS/MS, liquid chromatography-tandem mass spectrometry; NGS, next-generation sequencing; EV, extracellular vesicles; HMEV, human milk derived extracellular vesicles.

**Table 1 cells-11-02345-t001:** Selected hMEV molecules and their potential functions.

Type	Selected Molecule	Potential Function	References
Proteins	Tetraspanins, i.e., CD9, CD63, CD81	Well-accepted exosome biomarkers	[29,31,83]
		Participate in exosome biogenesis, cargo sorting, and membrane fusion	[32,33,34,84]
	Transforming growth facter-β (TGF-β)	An immunosuppressive molecule May induce CD4^+^CD25^+^FoxP3^+^Treg differentiation from PBMCs	[29,83] [85]
	Intracellular adhesion molecule 1 (ICAM-1)	A cell adhesion molecule Play a role in EV trafficking Can bind to and modulate HIV-1 infection	[83] [86] [87]
	Integrins	Cell adhesion molecules Play roles in exosome intra- and extravasation and recipient cell targeting	[29,83] [88,89]
	Lactadherin (Milk Fat Globule-EGF-factor VIII; MFGE8)	A specific hMEV marker	[29,83,90,91]
		Protects against rotavirus infection	[92]
		Modulates bladder cancer development	[93]
	Alix	A well-accepted exosome biomarker Plays roles in the ESCRT pathway and EV biogenesis	[31,83] [32,33,34]
	Tumor susceptibility gene 101 (Tsg101)	A well-accepted exosome biomarker Plays roles in the ESCRT pathway and EV biogenesis	[31,83] [32,33,34]
	MHC class I and class II	Antigen presenting molecules Plays a crucial role in the adaptive immunity with antigen presenting potential	[29,83] [94]
	Butyrophilin	A specific hMEV marker	[29,83,90,91]
		Plays an important role in lactation and regulates secretion of milk lipid droplets	[95]
		Modulates T cell activation, induces Treg differentiation, and promotes γδ T cell development	[96]
	Heat shock protein 70 (HSP70)	A commonly used exosome marker	[29,31,83]
		Serves as the molecular chaperone to prevent protein aggregation and cellular stress	[97]
		May enhance anti-cancer immunity in colon cancer and melanoma	[98]
	Xanthine dehydrogenase (XDH)	A specific hMEV marker	[29,83,90,91]
		Binds to the cytoplasmic tail of butyrophilin and catalyze purine oxidation, plays roles in purine catabolism and production of ROS and NO	[99]
	Mucin-1	A major mucin glycoprotein expressed on the apical surface of mammary epithelial cells Binds DC-SIGN to block HIV-1 viral transfer from monocyte-derived dendritic cells to CD4^+^T cells Suppresses Toll-like receptor signaling (i.e., TLR2, 3, 4, 5, 7, 9) and regulates inflammatory responses to infection	[29,76,83] [100] [101,102]
	Flotillin-1	A lipid raft associated protein Regulates exosome release	[83] [54]
	Tissue factor	A transmembrane protein with procoagulant activity May prevent bleeding of maternal nipple skin damage and infant gastrointestinal vascular damage	[83] [8]
Lipids	Phospholipids	Intrinsic lipid components of EV membrane May support neurocognitive and pulmonary development in infants	[75]
	Sphingolipids	Intrinsic lipid components of EV membrane Involves in ESCRT-independent exosome biogenesis	[75] [50]
	Cholesterol	Intrinsic lipid components of EV membrane Maintains the stability of phospholipid bilayer of exosome membrane	[75]
Nucleic Acids	miR-148a-3p	The most abundant miRNA in hMEVs	[103,104]
		Modulates DNMT1 dependent DNA methylation	[105,106,107]
		Suppresses the progression of breast, pancreatic, gastric, bladder, cervical cancers, and Hodgkin lymphoma	[106,107,108,109,110,111]
		Suppresses p53 expression and mitigate NF-κB induced intestinal cell inflammation and apoptosis	[112]
		Modulate angiogenesis	[113,114]
		Exerts neuroprotection, promotes neural cell proliferation, and may involve in neurodevelopment and cognitive functions	[115,116,117,118,119]
	miR-30a/d-5p	High abundance hMEV-miRs Modulates cell proliferation and apoptosis	[103,120] [121,122]
	let-7a/b/f-5p	High abundance hMEV-miRs Regulates inflammatory processes linked to vascular function and neurological outcomes	[103,104] [123]
	miR-22-3p	A high abundance hMEV-miR Regulates cell proliferation and apoptosis Modulates gluconeogenic pathway through TCF7	[103,104] [124] [125]
	miR-146b-5p	A high abundance hMEV-miR	[103,126]
		Suppresses the development and progression of hematologic malignancies, i.e., T-ALL, B-ALL, and AML	[127,128]
	miR-200a/c-3p	A high abundance hMEV-miR	[103,129]
		Supports neuronal survival against amyloid-beta-induced ER stress and neurotoxicity	[130,131]
	LINC00657 (NORAD)	A high abundance lncRNA in hMEVs Maintains genome stability and regulate DNA repair	[80]

Abbreviations: AML, acute myeloid leukemia; B-ALL, B cell-acute lymphoblastic leukemia; ESCRT, endosomal sorting complex required for transport; MHC, major histocompatibility complex; HMEV, human milk-derived extracellular vesicle; HMEV-miR, HMEV-derived microRNA; lncRNA, long non-coding RNA; NORAD; non-coding RNA activated at DNA damage; miRNA, microRNA; ncRNA, non-coding RNA; NO, nitric oxide; ROS, reactive oxygen species; T-ALL, T cell-acute lymphoblastic leukemia; Tcf7, Transcription factor 7; Treg, T regulatory cells.

**Table 2 cells-11-02345-t002:** HMEV functions.

Action	Biologic/Therapeutic Effects of hMEVs	References
Gut maturation	Promote the proliferation of normal fetal colon epithelial cells, but not colon cancer cells, in a miR-148a-3p dependent manner	[133]
	Enhance gingival re-epithelialization via p38 MAPK mediated cell migration and cytoskeletal remodeling	[149]
Mitigate intestinal damage	Promote intestinal epithelial cell viability under H_2_O_2_ induced oxidative stress	[150]
	Promote intestinal stem cell viability under H_2_O_2_ induced oxidative stress via Wnt/β-catenin signaling	[151]
	Prevent LPS-induced epithelial cell injury in intestinal organoids and mitigate mucosal injury in an NEC model in vivo	[152,153]
Immunomodulation	Induce FoxP3 expression and promote CD4+CD25+FoxP3+ Treg differentiation	[29]
	Suppress activation and differentiation of CD4^+^CD45RA^+^ naïve T cells toward CD4^+^CD45RO^+^ memory T cells	[149]
	Suppress cytokine production: IFN-γ, IL-5, IL-9, IL10, IL13, IL-17, IL-22	[149]
	Inhibit expression and function of endosomal TLR3 supporting mucosal colonization of commensal bacteria in the newborn	[149]
	Reduce expression of inflammatory cytokines, i.e., IL-6 and TNF-α in colitis mouse colon	[154]
Attenuate viral infection	Competitively bind to DC-SIGN on monocyte-derived dendritic cells and inhibit HIV-1 viral transfer to CD4^+^T cells	[100]
	Interfere with early steps of rotavirus and respiratory syncytial virus replication	[132]
	Inhibit replication and cell attachment of human cytomegalovirus	[155]
	Inhibit Zika virus and Usutu virus	[156]

Abbreviations: DC-SIGN, dendritic cell-specific intercellular adhesion molecule-3-grabbing non-integrin; HIV, human immunodeficiency virus; IFN, interferon; IL, interleukin; LPS, lipopolysaccharide; NEC, necrotizing enterocolitis; TLR, Toll-like receptor; TNF, tissue necrosis factor; Treg, T regulatory cells.

**Table 3 cells-11-02345-t003:** Influence of maternal conditions on hMEV composition and potential health outcomes in children.

Maternal Condition	HMEVs			References
	Biological Change	Significant Molecular Markers	Functional Association	
Allergic sensitivity	Levels of hMEV-derived proteins in the milk of sensitized mother relative to nonsensitized	*Increase*: HLA-ABC *Decrease*: Mucin-1	Potentially influence the development of allergy in children	[76]
Diabetes type 1	HMEV-miRs differentially express in mothers with type I diabetes relative to healthy controls	*Increase*: miR-4497, miR-3178, miR-1246, miR-133a-3p, miR-1290, miR-320d *Decrease*: miR-518e-3p, miR-629–3p, miR-200c-5p	Induce expression of proinflammatory genes (IL1B, IL6, CXCL10, TNF-α) in human monocytes and modulate infant immune response	[77]
Gestational diabetes mellitus (GDM)	HMEV-miRs lower in milk of mothers with GDM	*Decrease:* miR-148a, miR-30b, let-7a, let-7d	Increase infant weight and fat mass	[78]
Overweight/obesity	HMEV-miRs lower in milk of overweight/obese mothers	*Decrease*: miR-148a, miR-30b	Increase infant weight and fat	[79]
Premature delivery	HMEV-derived lncRNAs differ in milk of mothers with preterm labor relative to term delivery	*Increase:* LRRC75A-AS1 *Decrease:* LINC00657 (NORAD), CTC-444N24.11, CRNDE	May induce adaptive responses of infants to prior hypoxic condition	[80]
	HMEV-derived peptides differ in milk of mothers with preterm birth relative to term delivery	*Increase:* 47 peptides, including bioactive peptides LTF (79–96) and HMGE8 (24–47) *Decrease:* 23 peptides	Promote intestinal cell proliferation and wound healing in vitro and prevent intestinal cell injury in a NEC mouse model	[81]
Stress/psychological distress	HMEV-miRs positively associated with degree of maternal stress during pregnancy	*Increase*: miR-99b-3p, miR-96-5p, miR-550a-5p, miR-616-5p, miR-155-5p, miR-604	Induce changes in fatty acid biosynthesis and metabolism, steroid biosynthesis, and the Hippo signaling pathway	[82]

Abbreviations: DM, diabetes mellitus; HLA, human leukocyte antigen; HMEV, human milk-derived extracellular vesicle; HMEV-miRs, HMEV-derived miRNAs; lncRNA, long non-coding RNA; miR, micro-RNA; MUC1, mucin-1; NORAD, non-coding RNA activated at DNA damage.

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
