# Peer review of "Human Milk Extracellular Vesicles: A Biological System with Clinical Implications"

_cells, 2022, doi:10.3390/cells11152345_

Round 1
Reviewer 1 Report
First, thanks for the opportunity. I liked the review, it contains important information for future works.
I have some suggestions:
1) FIG 1: on the X axis, the letter could be smaller to make the graph cleaner.
2) In lines 85 through 96, you are only using a reference. I suggest basing this information better.
3) Topic 2.4, line 235: When talking about works with SDS-PAge and others, which model was used. It is important to make it clear to the reader, as there are many articles gathered and each one must use a different model.
Ex: page 253, In vivo: in mice? Or humans?
To make the reading more fluid, I suggest you mention which model that method used.
4) About the mothers' physiological conditions, I really liked the presentation of the data, summarized, objective and clear, at least for me.
But always remembering about the source of that milk that was analyzed. They weren't all from human mothers.
Highlight this please.
Reviewer 2 Report
There are some critical points that should be revised:
Line 26: nomenclature: exosomes are a fraction of the big term “extracellular vesicles, EVs” and most of the studies dealt with EVs, not exosomes because of the wrong nomenclature. So, please remove “exosomes” from the entire manuscript text, unless you are sure that the isolated EVs are exosomes.
Line 32: Nobel prize in 2013 has been awarded to “vesicle trafficking” and that means many kinds of vesicles, but not only the EVs.
Line 59-61: Please revise the total number of published papers.
There is no mention of the specific isolation protocols of hMEVs, TEM characterization or Cryo-TEM characterization.
Because of variability in hMEVs (Line 298), how we could reach a quality-controlled grade for therapeutics?
Authors should update references with newly published similar reviews:
- Babaker, M.A.; Aljoud, F.A.; Alkhilaiwi, F.; Algarni, A.; Ahmed, A.; Khan, M.I.; Saadeldin, I.M.; Alzahrani, F.A. The Therapeutic Potential of Milk Extracellular Vesicles on Colorectal Cancer. Int. J. Mol. Sci. 2022, 23, 6812. https://doi.org/10.3390/ijms23126812
- Aqil, F.; Munagala, R.; Jeyabalan, J.; Agrawal, A.K.; Kyakulaga, A.-H.; Wilcher, S.A.; Gupta, R.C. Milk exosomes-natural nanoparticles for sirna delivery. Cancer Lett. 2019, 449, 186–195.
- del Pozo-Acebo, L.; de las Hazas, M.-C.L.; Tomé-Carneiro, J.; Gil-Cabrerizo, P.; San-Cristobal, R.; Busto, R.; García-Ruiz, A.; Dávalos, A. Bovine milk-derived exosomes as a drug delivery vehicle for mirna-based therapy. Int. J. Mol. Sci. 2021, 22, 1105.
- González-Sarrías, A.; Iglesias-Aguirre, C.E.; Cortés-Martín, A.; Vallejo, F.; Cattivelli, A.; del Pozo-Acebo, L.; del Saz, A.; de las Hazas, M.C.L.; Dávalos, A.; Espín, J.C. Milk-derived exosomes as nanocarriers to deliver curcumin and resveratrol in breast tissue and enhance their anticancer activity. Int. J. Mol. Sci. 2022, 23, 2860.
Reviewer 3 Report
I have read with great interest the review paper entitled “Human Milk Extracellular Vesicles: A Biological System with Clinical Implications” by Chutipongtanate et al. The manuscript summarizes a significant piece of evidence regarding human milk-derived extracellular vesicles (hMEVs) reviewing their role in health and disease and highlighting their potential clinical effects. Generally speaking, the review is well written and organized. It is also remarkable that authors have addressed the bioavailability of hMEVs. I only have some minor suggestions:
- The novelty of the review is hampered by the fact that other reviews on the same topic have recently been published (PMID:34681274; 34249944; 32414045).
- Page 2, line 36: cardiology (PMID: 35681540) should also be added as a field of knowledge.
- The effects of hMEVs in coagulation should be added (PMID: 35748324; 33351123). Other biological functions such as immunometabolism as well.
- Table 1 should be organised by molecular components (proteins, lipids, nucleic acids).
- References of Table 1 refer to the potential function of each component not to the studies where they were identified, which should be incorporated for completeness.
- Table 1 lacks tissue factor for instance. Please revise.
Round 2
Reviewer 2 Report
The authors addressed my comments.